# Trutinor: a Conceptual Study for a Next-Generation Earth Radiant Energy Instrument

**Cindy L. Young** [1,*]**, Constantine Lukashin** [1]**, Patrick C. Taylor** [1]**, Rand Swanson** [2]**, William S. Kirk** [2]**, Michael Cooney** [1]**, William H. Swartz** [3]**, Arnold Goldberg** [3]**, Thomas Stone** [4]**, Trevor Jackson** [1]**, David R. Doelling** [1]**, Joseph A. Shaw** [5] **and Christine Buleri** [6]

1 NASA Langley Research Center, Hampton, VA 23666, USA; constantine.lukashin-1@nasa.gov (C.L.); patrick.c.taylor@nasa.gov (P.C.T.); michael.p.cooney@nasa.gov (M.C.); trevor.p.jackson@nasa.gov (T.J.); david.r.doelling@nasa.gov (D.R.D.)
2 Resonon Inc., Bozeman, MT 59715, USA; swanson@resonon.com (R.S.); kirk@resonon.com (W.S.K.)
3 Johns Hopkins University Applied Physics Laboratory, Laurel, MD 20723, USA; bill.swartz@jhuapl.edu (W.H.S.); arnold.goldberg@jhuapl.edu (A.G.)
4 United States Geological Survey, Flagstaff, AZ 86001, USA; tstone@usgs.gov
5 Department of Electrical & Computer Engineering, Montana State University, Bozeman, MT 59717, USA; joseph.shaw@montana.edu
6 Quartus Engineering Inc., San Diego, CA 92121, USA; christine.buleri@quartus.com
* Correspondence: cindy.l.young@nasa.gov; Tel.: +1-757-864-7622

**Abstract:** Uninterrupted and overlapping satellite instrument measurements of Earth's radiation budget from space are required to sufficiently monitor the planet's changing climate, detect trends in key climate variables, constrain climate models, and quantify climate feedbacks. The Clouds and Earth's Radiant Energy System (CERES) instruments are currently making these vital measurements for the scientific community and society, but with modern technologies, there are more efficient and cost-effective alternatives to the CERES implementation. We present a compact radiometer concept, Trutinor (meaning "balance" in Latin), with two broadband channels, shortwave (0.2–3 μm) and longwave (5–50 μm), capable of continuing the CERES record by flying in formation with an existing imager on another satellite platform. The instrument uses a three-mirror off-axis anastigmat telescope as the front optics to image these broadband radiances onto a microbolometer array coated with gold black, providing the required performance across the full spectral range. Each pixel of the sensor has a field of view of 0.6°, which was chosen so the shortwave band can be efficiently calibrated using the Moon as an on-orbit light source with the same angular extent, thereby reducing mass and improving measurement accuracy, towards the goal of a gap-tolerant observing system. The longwave band will utilize compact blackbodies with phase-change cells for an absolute calibration reference, establishing a clear path for SI-traceability. Trutinor's instrument breadboard has been designed and is currently being built and tested.

**Keywords:** earth radiation budget; climate change; small satellite constellation; microbolometer array; phase change cells; carbon nanotubes; lunar calibration; CERES; ARCSTONE; RAVAN

## 1. Introduction

Climate change has the capacity to alter many aspects of the Earth system, as well as reduce economic stability, impact national infrastructure and security, and affect daily life. A critical aspect of monitoring and understanding climate change is maintaining a continuous record of Earth's radiation budget (ERB). The radiation budget is the balance between absorbed solar radiation and emitted thermal infrared radiation by the Earth–atmosphere system. The Clouds and Earth's Radiant Energy

System (CERES) instruments currently collect this data [1,2]. It takes decades to confidently detect climate-relevant trends, and the scientific and societal value of the current record will significantly degrade if there is a gap in observations. The CERES instrumentation was developed in the 1990s based on Earth Radiation Budget Experiment (ERBE)-scanner instrument heritage [1,2]. Continuing the critical ERB measurements with old technology is impractical and financially unsustainable. Our goal is to outline a long-term, cost-effective, and sustainable solution employing current and emerging technologies in a compact and cost-effective radiometer.

CERES has been a successful approach, but it has design features that can be improved to reduce mass, cost, and measurement uncertainty, increasing the efficiency of future missions. For example, the traditional hosted payload approach requires a large pedestal for the required field of regard and two-dimensional gimbal to enable data collection across different viewing geometries to sample the anisotropy of the radiation field. The CERES approach for observations, using a single field-of-view radiometer, requires continuous scanning across the field of regard, increasing instrument mass and power use requirements to obtain Earth-view sampling. It also must be hosted on a larger satellite with an imager, which determines if the CERES instrument is observing clouds, aerosols, or clear skies. The scene type classification is very important and allows for the derivation of cloud and aerosol forcing, major contributors to uncertainty in climate predictions [3]. A compact sensor integrated into a small satellite bus, relying on satellite pointing ability and flying in constellation with an existing imager, can result in an instrument concept with a smaller size and lower mass and power requirements. Another issue with the current system is the insufficient measurement accuracy to tolerate a gap in the data record, which can be mitigated by improving the accuracy of the instrument's on-orbit calibration. We focus on cost-effective implementations of a new instrument design and mission architecture.

We present a concept for a compact radiometer: the Trutinor instrument (trutinor means "balance" in Latin), integrated into a small satellite, flying in formation with an existing imager on another satellite platform. The concept incorporates very useful technologies developed under two projects funded by the NASA Earth Science Technology Office (ESTO): (1) ARCSTONE: Calibration of Lunar Reflectance from Space—this project is already in development to establish the Moon as an on-orbit shortwave high-accuracy calibration source, removing the mass and risk of complicated onboard calibration; (2) RAVAN (Radiometer Assessment using Vertically Aligned Nanotubes): compact radiometer and black-body with carbon nanotube technology [4]. We have formulated science requirements and developed a novel instrument concept for making shortwave (SW) and longwave (LW) broadband observations of the Earth. The concept is based on a push-broom radiometer approach with a microbolometer detector array.

## 2. Science Motivation

Energy is the currency of our climate system. The temperature does not change, and rain does not fall, unless energy flows from place to place. The pathways and characteristics by which energy flows through the Earth system determines climate. The balance of solar radiation entering Earth's climate system, and both reflected solar and emitted terrestrial radiation leaving at the top-of-the-atmosphere (TOA) constitutes Earth's radiation budget (ERB). The difference between the amount of energy entering and exiting the climate system dictates the energy available to drive the global atmosphere–ocean circulation system, melt snow and ice, and raise the global temperature, making ERB the most fundamental constraint on climate. ERB modulates many societally-relevant aspects of climate, including global sea level, the mass of water tied up in glaciers and ice sheets, regional precipitation distributions and global mean precipitation, and global cloud patterns (e.g., [3,5]).

Continuity in the ERB record is paramount and represents an invaluable contribution to science and society. It enables the science community to address the most fundamental questions in climate science, as identified in the Earth Science and Applications from Space (ESAS) Decadal Survey [6] and a range of other relevant community reports and publications [7–11]. Moreover, a continuous, century-long ERB record is necessary to confidently answer the "most important" climate science

questions from the ESAS Decadal Survey [6] regarding climate sensitivity and cloud feedback. ERB measurements represent a fundamental constraint on the global hydrologic cycle and are also highlighted in the Decadal Survey as contributing to the "most important" and "very important" questions in hydrologic science. The continuity in the TOA ERB record also provides key data to assess, for example, how the energy budget for land and ocean domains and energy transport between land and ocean change with time, cross-equatorial energy transport by ocean and atmosphere that influence the location of the Intertropical Convergence Zone, and the hemispheric asymmetry of precipitation [12]. Such a multi-decadal ERB record has the potential to answer longstanding, fundamental questions in climate science; however, it has not yet been collected.

The CERES instruments [1,2] were first deployed on the NASA Tropical Rainfall Measuring Mission(TRMM) satellite in 1997 and are currently operating on NASA Terra and Aqua, Suomi-NPP, and NOAA-20 satellites (see Section 3 for more details). For the past 20 years, the CERES radiation budget science project has worked to create the first global, multi-decadal record of the ERB, as shown in Figure 1. The CERES ERB record has been the cornerstone of many climate sensitivity and cloud feedback investigations (e.g., [13–15]), as well as a key data source for climate model evaluation (e.g., [16–19]). These measurements and ERB science possess heritage, beginning with ERBE, and represent a culmination of extensive studies and lessons learned. Despite providing measurements of ERB beginning in the 1980s and answering critical questions (e.g., by directly measuring the global mean Earth albedo to be 0.29 [20,21]), the gap (i.e., loss of radiometric traceability) between the ERBE and CERES observations cannot be adequately resolved at the 0.2–0.3 W/m$^2$ per decade level required to analyze climate-relevant signals [12]. The transition from ERBE to CERES measurements demonstrates the consequences of a gap in an ERB data record without overlap and having inadequate accuracy. No new CERES instruments will be launched, and the CERES FM6 baseline mission ends in 2026 (NOAA-20 baseline lifetime). Thus, there is a significant probability of a gap in the planned ERB observations that requires urgent action, or the ability to tackle the most challenging, societally-relevant climate science questions, namely cloud feedback and climate sensitivity, is substantially degraded.

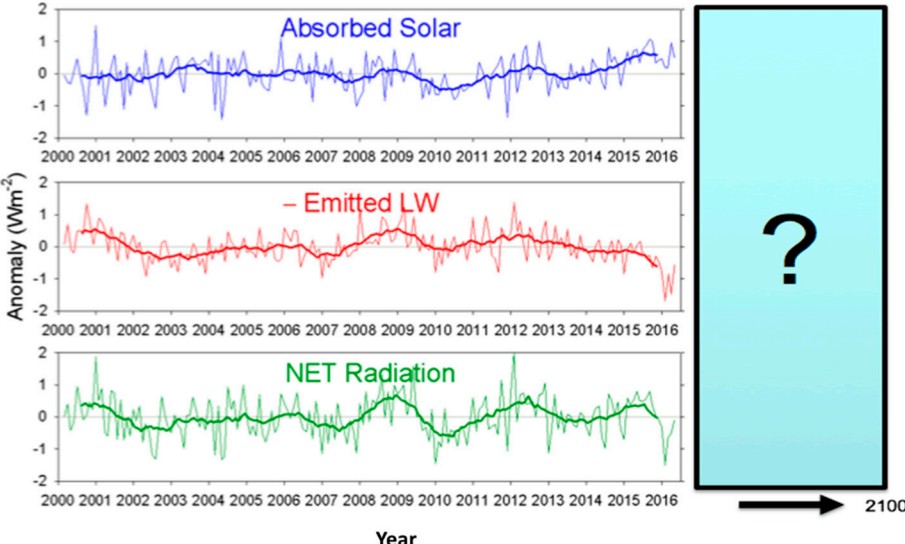

**Figure 1.** The 16-year top-of-the-atmosphere (TOA) Earth's radiation budget (ERB) record from Clouds and Earth's Radiant Energy System (CERES) for absorbed solar (blue), emitted longwave (red), and net radiation (green). The "?" represents the ~80 years of ERB data needed to extend the ERB record through the 21$^{st}$ century, and holds the secrets to the climate sensitivity of Earth, the magnitude of cloud feedback, the magnitude and controls on internal multi-decadal variability, and the radiative forcing of the climate. Plots courtesy of Norman Loeb.

The ERB record currently sits in a precarious position. According to the gap risk analysis, shown in Figure 2A,B, performed by the ERB Science Working Group Report [12], the likelihood of a gap in the ERB record exceeds 50% by as early as 2026, depending upon deorbit assumptions. CERES has been designed to achieve 1% and 0.5% accuracy in shortwave (SW) and longwave (LW) measurements, respectively. Even so, the CERES absolute accuracy is insufficient to capture climate relevant trends in the event of a gap in the ERB record; thus, a continuous ERB record requires overlap. Measurements with improved accuracy are needed to decrease the time for detecting trends in climate parameters and to reduce the impacts of a climate data gap [10,22].

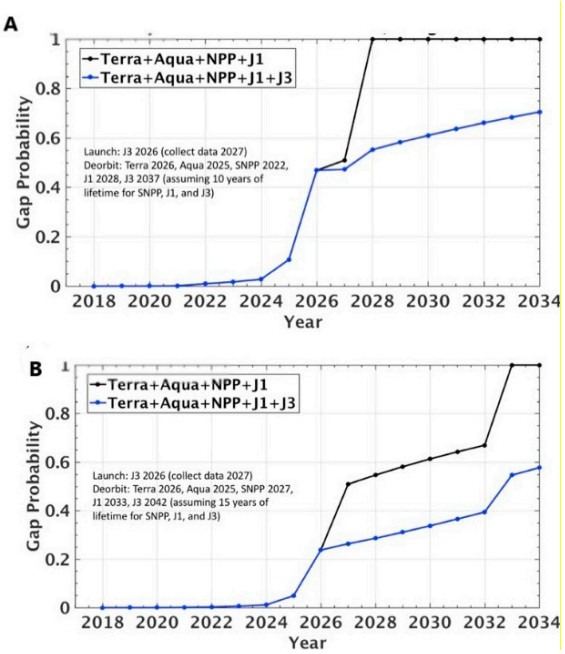

**Figure 2.** Analysis of the gap probability for (**A**) 10 years and (**B**) 15 years of mission lifetime for the Suomi National Polar-orbiting Partnership (SNPP), J1 and J3. Plots courtesy of Norman Loeb.

The final CERES FM6 instrument was launched aboard NOAA-20 in November 2017. The successor to CERES, the Radiation Budget Instrument (RBI), was canceled in January 2018, due to budgetary and schedule concerns. The RBI relied largely on a CERES-like approach. The Earth Venture Continuity (EVC-1) opportunity has recently been awarded to fund development of an instrument, Libera, capable of extending the CERES record (cost-capped at US$150 million ) and is intended as a short-term, rapid stop-gap solution for ERB continuity that gives the option to use RBI hardware [23]. This continuity mission is needed urgently, but the climate science community would greatly benefit from investigations of how to continue this measurement for decades on a budget of much less than US$150 million per instrument, a cost cap that may prove to be unsustainable. Reducing costs by about an order of magnitude would help to ensure that this valuable measurement continues for generations, and could enable multi-satellite observing systems capable of accomplishing additional science with more spatial coverage.

We contend that a continuous and long-term (>50 years) ERB data record will not be assured without a smaller and more cost-effective instrument. A new concept for ERB is urgently needed. The EVC-1 technology readiness level (TRL) requirements are too high (TRL 6 by Preliminary Design Review) to allow for serious consideration of future, more sustainable technologies. Trutinor represents (1) an innovative new ERB instrument utilizing cutting-edge technologies, (2) a potentially more accurate ERB instrument that will reduce both the time to detect trends in climate parameters and the impact of gaps in the ERB record, and (3) a cost-effective, compact instrument integrated onto a free-flying small satellite. The goal of Trutinor is to develop a novel, innovative, compact instrument

concept, extending the ERB record started under ERBE and CERES in a sustainable and cost-effective manner. The Trutinor vision is to enable a 50-year-long (and beyond), continuous ERB climate data record capable of resolving the key outstanding challenges in climate science (i.e., climate sensitivity, cloud feedback, and aerosol forcing), while providing a foundation for the next generation of climate scientists to pose new questions.

## 3. State of ERB Technology

For the last 20 years, there have been few significant efforts to advance technology relevant to ERB measurements. All relevant technologies, such as ERBE [24], Scanner for Radiation Budget (ScaRaB) [25], and CERES [2], were developed in the 1980s and 1990s. RAVAN, a 3U CubeSat for measuring Earth's radiation budget that launched in 2016, was one of the few recent technology developments, but it had a very large field of view that prevented resolution of aerosol and cloud radiative forcing [4]. ERBE was the first multi-satellite system designed to measure the Earth's radiation budget [24]. It included scanning and non-scanning instruments. ERBE scanners had three broadband channels (SW, LW, and total), the ability to scan in elevation, and a rather large ground footprint of 35 km cross-track and 45 km along-track. Radiometric channels viewed the Earth, internal calibration sources, and a solar-diffuser plate. The detectors were thermistors that used deep space observations on every scan as a reference point to guard against drift/offset, and they were located at the focal point of an f/1.84 Cassegrain telescope. Internal calibration included the Mirror Attenuator Mosaic (MAM) for solar observations, blackbodies, and lamps monitored by silicon photodiodes. The calibration radiometric accuracy for unfiltered radiances of the NOAA-9 ERBE scanner has been shown to be ±2.5%, ±1.2%, and ±1.8% in the shortwave, nighttime longwave, and daytime longwave, respectively [26].

The ScaRaB instrument was a four-channel (visible, SW, total, and window) cross-track scanning radiometer with a 40 km footprint at nadir. On the ScaRaB FM-3 flying on Megha-Tropiques, on-orbit calibration was achieved with a lamp, blackbody, and a set of filters. The Centre National d'Etudes Spatiales (CNES) budget computation demonstrated that SW radiance accuracy is less than 2% at 1-sigma and 1% in the LW domain [25]. However, it must be noted that all calibrations relied on a single source, which could potentially drift in-orbit. The major downside of the ScaRaB approach was that the instrument did not have an azimuth pointing degree of freedom and was not capable of collecting data for developing angular distribution models (ADMs), which are required to convert radiances into TOA fluxes. It relied on ADMs developed using CERES/TRMM data [27], which resulted in large uncertainties in TOA fluxes.

CERES is the most recently implemented broadband radiometer described in [2]. The CERES instrument is designed to provide a climate dataset suitable for examining the role of clouds in the radiative Earth–atmosphere system, and has significant ERBE heritage. CERES measures SW, LW, and total (LW + SW) radiance from the Earth using three Cassegrain telescopes, with a single precision thermistor bolometer detector for each channel. To use the total channel to calculate either SW or LW radiance, the field of view for all telescopes must be shared, and the telescopes need to be co-registered. Radiance is a measured quantity that depends upon the viewed scene and anisotropy. To measure the entire radiant energy flow within the Earth–atmosphere system, ADMs must be constructed from radiance measurements sampled at different viewing angles over different physically-defined scenes [28]. These ADMs take into account surface type and cloud physical and radiative properties derived from coincident imager observations, to compute radiative fluxes at the TOA. To develop ADMs, CERES instruments have a specific operation mode with a rotating azimuth plane to sample across the entire hemisphere of scattered and emitted broadband radiation. This scanning approach requires an azimuth gimbal that increases size, mass, power, and costs.

The CERES approach for converting measured radiance into TOA flux is advanced in comparison to ERBE and ScaRaB—it relies on data fusion with nearly coincident multi-spectral imager observations. Cloud properties and clear sky scenes are identified for CERES with the aid of near-simultaneous measurements by a multispectral imager flying on the same satellite platform. The necessity to

be hosted on a larger platform with an imager limits the cadence at which CERES instruments may be launched, and requires a large pedestal to mount the instrument for full field-of-regard viewing. However, the science data products do not require simultaneous imager observations, but near-coincident imaging (up to three minutes apart) would allow for a free-flier concept [12]. Clear sky measurements are needed to determine the cloud radiative effect, and this drives the radiometer footprint size requirements for CERES (20–25 km at nadir). We should clarify that these numbers represent the diameter of a circle of equivalent surface to the instrument static optical field-stop, shown as a hexagon in Figure 3. In reality, the CERES footprint is defined by the point spread function (PSF) [29] (CERES ATBD, Subsystems 1.0 and 4.4), which is as large as 63 km × 200 km at a 48° scan angle (see Figure 3C). A smaller footprint would provide more clear sky sampling, a major science requirement, as well as higher fidelity cloudy sky measurements, enabling smaller scale heterogeneities within clouds to be captured. The Trutinor instrument is designed with a smaller footprint size (~10 km at a 700–800 km altitude polar orbit) to enable innovative science.

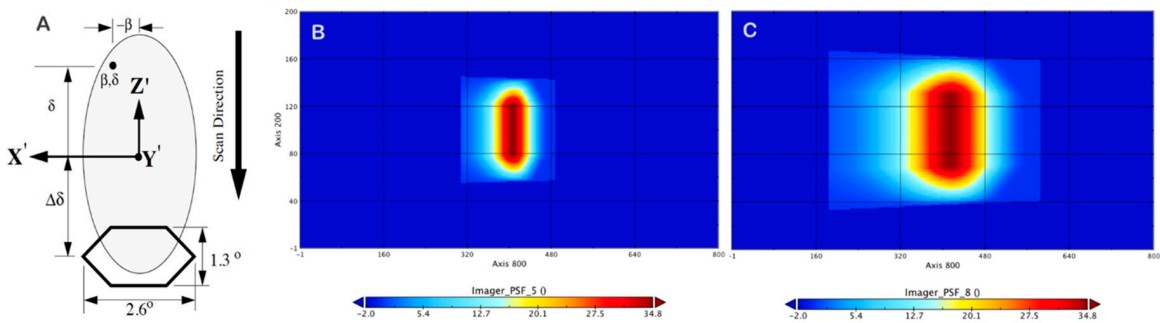

**Figure 3.** (**A**) Illustration of the CERES Point Spread Function from the ATBD; (**B**) and (**C**) Simulated CERES PSF for scan angle 29° and 48°, respectively. The edges of PSF are shown in light blue. Footprint size in along-scan direction changes from 90 km to 200 km.

CERES goals for calibration are 1% in SW and 0.5% in LW, achieved by combining ground and on-orbit calibrations. The on-orbit radiometric calibration system includes a blackbody for the total channel, and tungsten filament lamps and a mirror attenuator mosaic (MAM) for the SW channel. The SW channel of the CERES instruments on Terra and Aqua platforms experienced a degradation and exposed an inherent challenge for broadband radiometry—measuring only the broadband integrated signal while the incoming signal has varying spectral features. The CERES on-board calibration is not adequate to resolve this problem, because the tungsten filament lamps are not capable of monitoring the spectral response of the instrument in the UV/blue wavelength range. Additional issues include using a solar diffuser without a dedicated monitor for its degradation and referencing the lamps' light intensity to degrading silicon photodiodes.

All three instruments described above represent single-footprint scanning instruments hosted on a large space platform. This approach requires a massive pedestal to provide the required field of regard and heavy, power-hungry gimbals to enable the necessary scanning rate and azimuth rotation. Although the radiometric part of the instrument is not very large, the total package weight is 54 kg [30].

The Radiometer Assessment using Vertically Aligned Nanotubes (RAVAN) technology demonstration mission has successfully flown on a 3U CubeSat within NASA's In-Space Validation of Earth Science Technologies (InVEST) program [4]. RAVAN demonstrated a radiometer that is compact, low cost, and enabled by new technologies: a gallium fixed-point blackbody as a built-in calibration source, and a Vertically Aligned Carbon Nanotube (VACNT) radiometer absorber. RAVAN has a large 135° field of view to measure radiative flux at TOA directly, which results in a ground footprint of thousands of kilometers. The RAVAN approach, with sampling on such a large scale, does not allow for continuation of the existing ERB data record, which requires cloud-free observations on a 20 km spatial scale.

Existing and near future ERB instruments are designed around the concept of being one instrument among many on a large, expensive satellite bus. The advantages in the past were handing off many key requirements to the bus (e.g., power, station keeping, command, and data handling). With the commercialization, and subsequent price reduction, of small satellite buses providing competitive functionality, large-hosted payloads are no longer required. Small satellites of low cost are particularly useful for an ERB measurement to lower the gap risk by flying multiple, independent satellites while being able to incrementally improve the instrument over time using new technology, such as quantum detectors instead of current bolometric (thermal) detectors.

## 4. ERB Science and Measurement Requirements

Trutinor's science traceability matrix (STM) and requirements are shown in Figure 4. Trutinor's requirements are driven by ERB science objectives (not instrument heritage). In many cases, they are similar to those for CERES (e.g., spectral ranges for the SW and LW channels). However, we have relaxed the requirement for Type a (random) uncertainty because ADM noise dominates the total uncertainty in TOA fluxes [28]. Our objective in this project is to improve SW measurement accuracy (Type B uncertainty). The SW accuracy required to achieve a gap-tolerant ERB instrument was investigated by [10] and determined to be 0.15% (k = 1). This study used 10 years of CERES data between 2000–2010; a period that contained no significant El Niño or volcanic events, and, as a result, small interannual variability. It is likely that estimates of the magnitude of natural variability from this decade are underestimated, leading to a strict 0.15% (k = 1) gap-tolerant accuracy requirement. Moreover, recent studies have argued that the magnitude of multi-decadal variability in the cloud radiative effect is large enough to mislead us into concluding that the long-term cloud feedback has the opposite sign (e.g., [31,32]) when considering 30 years of data. Considering multi-decadal natural variability, these studies suggest that 0.3–0.5% (k = 1) accuracy may be sufficient to achieve a gap-tolerant ERB instrument using the framework described by [10]. Further investigation is needed to confirm if the 0.3–0.5% SW accuracy requirement is sufficient to make Trutinor a gap-tolerant ERB instrument.

| Science Questions | Observables | Parameters | Instrument Requirements | Mission Constraints |
|---|---|---|---|---|
| How is Earth's climate changing and what is the role aerosols and clouds? | Longwave radiances | Spectral Range | 5 - 50 microns | Minimum 3-year mission lifetime |
| | | Dynamic Range | 0 to 180 W/(m² sr) | ADM collection by agile spacecraft operations (azimuth rotation) |
| | | Type B uncertainty (accuracy) | of the larger: 0.50 W/(m² sr) or 0.5% (k = 1) | Free-flying small satellite with orbit maintenance |
| | | Type A uncertainty (precision) | of the larger: 2.0 W/(m2 sr) or 0.5% (k=1) | Flying with existing coincident imager (< 3 mins apart) 2D point accuracy < 0.2° (goal); < 0.05° threshold |
| How is climate forcing influenced by varying cloud and aerosol properties? | Shortwave radiances | Spectral Range | 0.2 - 3 microns | Ability to view earth's surface, moon, and sun |
| | | Dynamic Range | 0 to 425 W/(m² sr) | iFOV = 0.6° to view moon in one pixel |
| | | Type B uncertainty (accuracy) | larger of 0.50 W/(m² sr) or *0.5% (k=1) | gFOV < 25 km |
| | | Type A uncertainty (precision) | of the larger: 0.5 W/(m2 sr) or 0.5% (k=1) | Radiometric stability <0.15% (k = 1) threshold; <0.1% (k = 1) goal *75% (TBD) Earth coverage daily in nominal data collection mode |

**Figure 4.** The Trutinor Science Traceability Matrix. Note that to address the science questions, the observables (SW and LW radiances) must be converted to fluxes using Angular Distribution Models (ADMs). * denotes constraints to be investigated as part of a science study.

ERB measurements require a near-coincident imager with multiple spectral bands in order to determine scene type (cloud type, clear sky, ocean, sea ice, etc.). We suggest that an imager used for acquiring ERB measurements does not need to provide as much spectral information as, for example, the Moderate Resolution Imaging Spectroradiometer (MODIS) or the Visible Infrared Imaging

Radiometer (VIIRS), but only 5–8 specific bands that are important for discriminating clear sky from clouds [12]. Developing a compact imager, flying on the same platform as Trutinor, would allow flexibility in orbit selection and launch opportunity. Since the need to be in a ~700–800 km orbit is primarily because this is where existing imagers are located, a co-manifested imager would free us to fly in a lower orbit, which would be easier and less costly to reach.

It should be noted that Trutinor's science objectives explicitly imply that new measurements will be compatible with the CERES data production system and algorithms, starting with the Level-2 products. The Trutinor Level-1B calibrated and geolocated data product will need to be prepared for being ingested into the CERES processing system.

## 5. The Trutinor Concept and Discussion

The Trutinor concept leverages available technology and detector arrays to sample Earth without heavy mechanical parts (Figure 5). The envisioned Trutinor instrument is a push-broom imaging radiometer with two broadband channels, one labeled SW (0.2–3 μm) and the other labeled LW (5–50 μm). The instrument uses a three mirror off-axis anastigmat telescope as the front optics [33,34] to image the two bands onto a microbolometer array coated with gold black, which provides good performance across the full spectral range. A ray-trace diagram for both channels of the Trutinor instrument is shown in Figure 5B,C. The blue and green colors indicate SW and LW channels, with roll (or elevation) angles ranging from 0° (nadir) to ±55°. Each pixel of the sensor has a field of view of 0.60°, which was chosen so the SW band can be calibrated using the Moon as a known source with similar angular extent, thereby designing calibration capability into the system. The LW calibration plan follows the strategy proposed for the Climate Absolute Radiance and Refractivity Observatory (CLARREO) [10], in which multiple phase change cells with cavity emitters were needed for temperature accuracy. Different phase-change materials cover a range of temperatures: mercury (−39 °C), water (0 °C), and gallium (30 °C). Compact gallium phase-change cells, for example, were used on RAVAN to monitor radiometer stability and can provide reliable LW SI-traceability on-orbit.

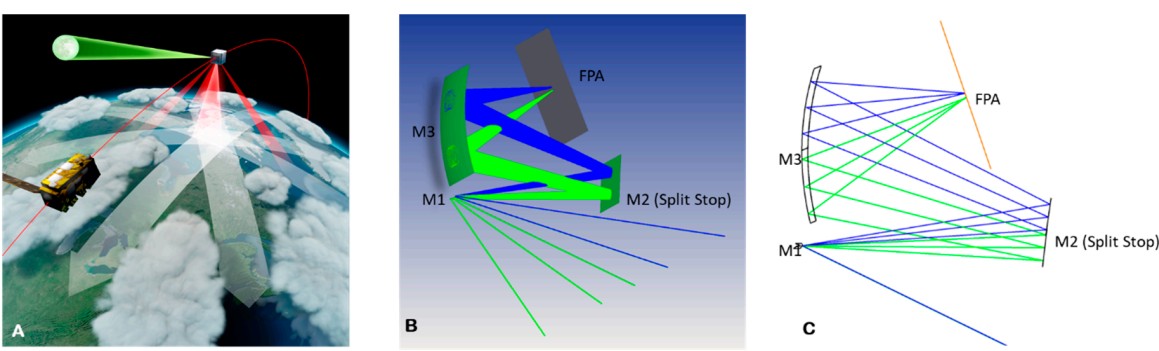

**Figure 5.** (**A**) Trutinor observatory, flying in constellation with an imager, performing lunar calibration and collecting multi-angular measurements; (**B**) and (**C**) 3D and 2D ray-trace representations of Trutinor optical design, respectively.

The Trutinor design is motivated by a practical approach for accurate calibration of the SW channel using the lunar irradiance [35,36]. The instrument pixel field of view is fixed to 0.6°, just slightly larger than the 0.52° angular size of the full Moon, to account for pointing uncertainties. The desired total instrument field of regard is 110°, which requires 184 pixels. This is readily achieved with microbolometer arrays. The science requirements dictate a nadir footprint of 25 km or less. With a 0.6° field of view (10.5 mrad), this means the instrument can be in orbit at any altitude less than 2400 km, which is consistent with low Earth orbiting satellites. Thus, from an imaging standpoint, the calibration requirement aligns well with mission science goals from the anticipated platform.

The concept images the two channels onto different rows of the microbolometer array. This can be done by utilizing a split stop layout, as shown in the ray-trace diagram in Figure 5C. The optical layout consists of (1) an off-axis biconic mirror (M1) with two different curvatures to accommodate a field of view of 110° by 0.6°, which reorients and contracts the entrance light, (2) a second mirror (M2), having two flat surfaces with a 1° angular difference in orientation, which acts as a split stop to redirect the two beams to the focusing mirror, and (3) an off-axis biconic mirror (M3) that focuses the two beams onto the focal plane. Thus, the input signal is divided into two channels by imaging the signal onto two rows of the microbolometer array, which are illustrated as blue rays for one channel and green for the other (Figure 5B,C). Filters for the two channels are not shown. Figure 5C shows nadir-viewing rays, while Figure 5B includes other field angles. Suitable mirror coatings will be based on heritage instruments, such as: Earth Clouds, Aerosols, and Radiation Explorer (EarthCARE), CERES, ScaRaB, etc. It appears that the most robust (space rated) aluminum (Al)-coated mirrors have a similar UV spectral response to silver (Ag)-coated mirrors. However, Ag-coated mirrors, in general, have a higher and more uniform spectral response in the near ultraviolet (NUV) and visible–near-infrared (vis–NIR) wavelength regions and a more predictable spectral response over time (i.e., the coatings are more robust).

The selected sensor is a microbolometer array from Institut National d'Optique (INO). A detector with 70 μm pixels makes imaging of all Earth-reflected and -emitted radiation feasible, although challenging for the longest wavelengths. Spot-diagrams from preliminary ray-trace modeling (not shown) indicate the signal images within the 70 μm pixels at all field angles. The bolometer array is similar to a sensor INO developed for the EarthCARE project [37], leveraging lessons learned from that experience.

A concern when using a microbolometer is achieving a sufficient signal-to-noise ratio (SNR). This is somewhat compounded by the fact the preliminary optical design has a relatively large f/# at f/5. However, assuming a typical specific detectivity (D*) of 108 (cm $\sqrt{Hz}$/W), an input signal of 100 W, 50% losses, and a 1-second allowed viewing time, one calculates an SNR of 155. This result is slightly lower than desired, and suggests additional work will need to be put into the optical design to decrease the f/#. This is possible, as the optical design has not yet included higher-order terms for all surfaces. It is also worth noting that free-form optic fabrication equipment can readily accommodate aspherical curvatures. The important point to note is that the modeled SNR is reasonable as is, and there is a clear pathway to improvements with additional design work.

Next we describe two development cycles needed to increase the TRL of Trutinor subsystems. The first is the Trutinor Breadboard, which is an ongoing project designed to demonstrate the optical concept and detector. The Trutinor engineering design unit (EDU) for the entire instrument would follow.

Trutinor Breadboard: the breadboard instrument design (configuration diagram shown in Figure 6A) maximizes the use of commercial off-the-shelf (COTS) components, including a COTS microbolometer array from INO (Microxcam-384I-MLWIR, 384x288 pixels, 3–14 μm bandpass), to illustrate the measurement technique in an accelerated and cost-effective manner. The breadboard build and alignment is being conducted by Resonon at their facility in Bozeman, MT, serving as a proving ground for the instrument concept. Characterization of the breadboard instrument at Resonon, with support from Montana State University, will achieve TRL 3 for Trutinor.

Trutinor EDU: Future work is needed to construct the engineering design unit (EDU) instrument (configuration diagram shown in Figure 6B), which will be designed to meet the science requirements, as well as the requirements for environmental testing, resulting in an instrument that is closer to a flight prototype instrument than a laboratory-only instrument. The EDU design and calibration approach is described below. The optical system will utilize a split beam design for two focus locations on a single custom microbolometer array. The two focus locations will be referred to as the SW focus (0.2–3 μm) and the LW focus (5–50 μm), with corresponding filters on the microbolometer array for each wavelength range. For LW calibration, the EDU instrument will utilize a two-dimensional (flat)

carbon nanotube black body or cavity emitter with integrated phase-change cells [4] for absolute calibration of the microbolometer array in a configuration similar to what will later be used for flight. To monitor SW relative spectral response (RSR) in-orbit, provisions will be made for a narrow band filter sub-system, with at least two bands (in blue and red wavelengths), to be placed at an appropriate location at the entrance of the optical system to monitor its degradation, again in a configuration similar to what will be used for flight. Trutinor's RSR monitoring plan is to track the signal ratio of the blue and red filters from the beginning of the mission, using two filters for each band at short- and long-time intervals. A thermal stability and temperature measurement sub-system will be included in the instrument design to warm bias the instrument and achieve thermal stability of approximately a tenth of a degree via heaters.

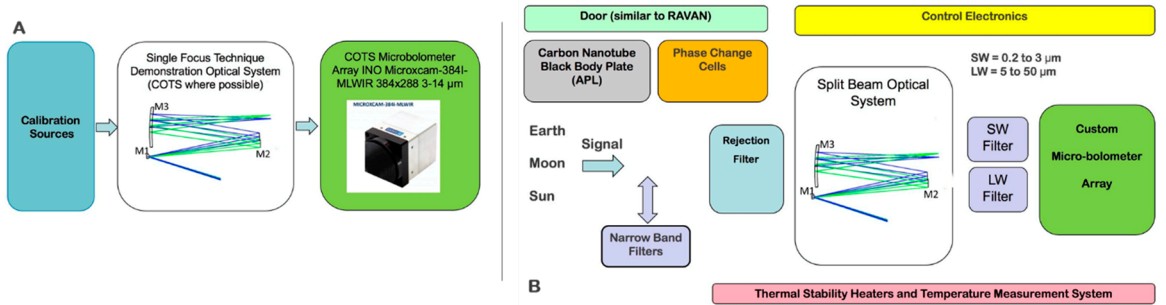

**Figure 6.** (**A**) the Trutinor Breadboard diagram; (**B**) the Trutinor EDU system block diagram.

Shortwave Calibration: Trutinor's SW channel will be calibrated in orbit against the Moon. This novel approach eliminates on-board calibration hardware, saving mass, complexity, and instrument development time. Our calibration accuracy goals will be achieved by using a high-accuracy lunar radiometric reference that is currently under development through efforts of NASA's ARCSTONE project [38], NIST's Ground- and air-LUnar Spectral Irradiance (LUSI) [39], and USGS' RObotic Lunar Observatory (ROLO) lunar model [40–42], which is expected to be available before a space deployment of Trutinor. Using the Moon for absolute radiometric calibration makes Trutinor unique among Earth-observing sensors. Each of Trutinor's 184 detector pixels will be exposed to signal from the entire lunar disc, and will use lunar broadband irradiance as an absolute calibration reference. The large number of readings per detector will reduce random uncertainty. Illumination of all 184 detectors will be achieved by satellite pointing and tracking the location of the Moon.

To accommodate the SW sensor's broadband response, the spectral range of the lunar calibration reference must be expanded from the current 0.35 μm UV limit to 0.2 μm, and from the current 2.3 μm infrared limit to 3.0 μm. Extension of the UV limit is relatively straightforward, requiring only appending the current lunar reflectance spectrum in the lunar calibration system with supplemental data that cover the extended wavelength range. Whole-disk lunar reflectance spectra in the UV acquired by the SOLar-Stellar Irradiance Comparison Experiment (SOLSTICE) on the SOlar Radiation and Climate Experiment (SORCE) mission [43] can provide this extended range.

Extending the lunar reference to 3.0 μm in the infrared requires accounting for the thermal signal emitted by the sunlit Moon, although this is a very small part of the total lunar signal spectrally integrated from 0.2 to 3.0 μm [44]. Quantifying the thermal part of the lunar irradiance involves developing a model for the temperature distribution over the hemisphere of the Moon seen by an observer, coupled with a thermal radiance model appropriate for the lunar surface (e.g., [45]). The lunar surface reaches nearly 400 K at the sub-solar point, falling off toward the terminator with a cosine-power functional dependence [46]. The Trutinor team has a path forward to extend the ARCSTONE effort (lunar calibration from 350 to 2300 nm), and to establish the Moon as an accurate SW broadband and SI-traceable reference through future modeling studies.

Longwave Calibration: an accurate and reliable calibration converting the output digital data of the sensors into source radiance or temperature is essential to meet the ERB science goals. The strategy to be employed will include extensive calibration of the detectors on the ground, and the use of highly stable blackbody calibration sources for the LWIR part of the instrument. One possible strategy that could be employed for Trutinor is that being implemented on the Compact Infrared Radiometer in Space (CIRiS) instrument. The CIRiS instrument uses a 640 × 480-pixel microbolometer imaging array, and the calibration sources are two carbon nanotube (CNT)-coated extended-area blackbody radiators whose temperatures can be varied independently from −40 °C to +70 °C. The emissivity of the carbon nanotube emitters is >0.996, which reduces the systematic errors from the uncertainty of the emissivity and reduces stray light caused by reflections off the radiator surfaces. Sensor response to calibrated blackbody sources has traditionally been the primary method used for calibrating IR sensors. The photon and power emission spectra of an ideal blackbody is precisely known from the Planck radiation law, and blackbody sources that closely approach the theoretical values of radiance in both cavity and extended-area formats have been produced commercially. Essential to getting sufficient accuracy of the source temperature derived from the sensor will be the accuracy of the temperature of the blackbody radiator, as well as the stability of the temperature, over the time during which calibration measurements are made. In addition, we will need very precise knowledge of the emissivity spectrum of the radiating surface of the calibration blackbody. Carbon nanotubes are known to have very high (>0.996) emissivities and are able to warm up and cool down rapidly, compared to other blackbody radiator materials. Therefore, we plan to use a CNT-coated radiator surface. Carbon nanotubes have been demonstrated both as radiometer absorbers and black body emitters on the RAVAN CubeSat mission [4]. In addition, phase-change cells will be needed for temperature accuracy. A mechanism to move various calibration sources into the Trutinor field of view will be developed for the EDU. It will also be very important to obtain the precise spectral response of the detectors and the transmission and/or reflectance spectra of the various elements (mirrors, lenses, filters, etc.) in the optical system to ensure an accurate calibration.

## 6. Conclusions

Decades of continuous ERB data are required to determine statistically-significant climate-relevant trends, enabling the quantification of cloud feedback and climate sensitivity. A loss in radiometric traceability in the ERB record will create a time gap and increase the uncertainty of the analyzed trends. As a result, our ability to quantify cloud feedback and climate sensitivity will be severely hampered, potentially delaying this knowledge by a decade or more. This major impact of a gap manifests because the absolute accuracy of CERES is insufficient to capture climate-relevant trends (~0.2–0.3 Wm$^{-2}$ per decade). Recent events demonstrate that the current CERES instrument hosted-on-payload approach is unsustainable, and presents a high risk for the long-term continuation of ERB measurements. The ERB record sits in a precarious position, with a gap possible as early as 2026. The EVC-1 instrument, Libera, is intended to be a one-off solution, as it is a scanning radiometer similar to CERES, hosted on the JPSS-3 satellite. Development of a smaller, more cost-effective, and more accurate ERB instrument is necessary to substantially reduce the risk to achieving continuity in this vital, societally-relevant climate data record. The ultimate solution for ensuring continuity is the creation of a gap-tolerant instrument, which requires high absolute accuracy (between 0.15 and 0.5%).

Trutinor is a concept for a highly accurate, compact instrument for extending the ERB record started under ERBE and CERES in a sustainable and cost-effective manner. While CERES instruments require a scanning approach with an azimuth gimbal that increases size, mass, power, and cost, the Trutinor design offers a non-scanning solution onboard a free-flying small satellite. Since the science data products require near-coincident (up to three minutes apart), not simultaneous imager observations, a free-flier concept meets the science requirements and allows for more flexibility in launch dates. The Trutinor instrument design contains a smaller footprint size (~10 km) than CERES, to enable innovative science. A smaller footprint would provide more clear sky sampling, as well as higher

fidelity in cloud measurements, enabling smaller-scale heterogeneities within clouds to be captured. The higher spatial resolution created by a smaller footprint also ensures backwards compatibility with older instruments by averaging footprints and degrading spatial resolution. CERES goals for calibration are 1% in SW and 0.5% in LW, achieved by combining ground and on-orbit calibrations. Trutinor will achieve higher accuracy through using the Moon as a SW calibration source and using a narrow band filter subsystem to monitor changes in the SW RSR. Trutinor's requirements are driven by ERB science objectives (not instrument heritage). In many cases, they are similar to those for CERES, and Trutinor's science objectives explicitly imply that new measurements are compatible with the CERES data production system and algorithms. However, we have relaxed the requirement for Type a (random) uncertainty, because ADM noise will dominate the total uncertainty in TOA fluxes and aim to improve SW measurement accuracy (Type B uncertainty) by a factor of two. Studies considering multi-decadal natural variability suggest that 0.3–0.5% accuracy may be sufficient to achieve a gap-tolerant ERB instrument. Further investigation is needed to confirm if the 0.5% SW accuracy requirement is sufficient to make Trutinor a gap-tolerant ERB instrument.

We have refined science requirements and developed a novel instrument concept for continuing and improving the ERB record. The Trutinor instrument is a push-broom imaging radiometer with two broadband channels, SW (0.2–3 μm) and LW (5–50 μm). The instrument uses a three mirror off-axis anastigmat telescope as the front optics to image the two bands onto an INO microbolometer array coated with gold black, which provides good performance across the full spectral range. The concept is to image the two channels onto different rows of the microbolometer array, allowing the instrument to use the same optical train for both channels. This can be done by utilizing a split stop layout. The second mirror in the system is split into two sections to separate the signal. This divides the input signal into two channels by imaging the signal onto two rows of the microbolometer array. Each pixel of the sensor has a field of view of 0.6°, which was chosen so the SW band can be calibrated using the Moon as a known on-orbit reference source, thereby designing calibration capability into the system. The LW band will utilize compact blackbody sources, including phase-change cells for calibration, as has been demonstrated by the RAVAN mission. Trutinor's instrument breadboard is currently being built and tested.

The team has defined short- and long-term objectives to obtain an accurate Earth radiation budget record. The immediate 10-year plan leverages existing cutting-edge technologies, while the 30-year (and beyond) plan involves design cycling and updating as newer technologies become available. This will enable continued measurements of Earth's radiation system and archival of more accurate long-term flux measurements for generations, improving our understanding of aerosol and cloud impacts on climate.

**Author Contributions:** All the authors made significant contributions to the work. Conceptualization: C.L. (2015–present), C.L.Y., M.C., R.S. (2017–present), W.S.K., P.C.T., W.H.S., A.G., T.J., and D.R.D. (2019–present); methodology: C.L.Y., C.L., M.C., R.S., P.C.T., W.H.S., A.G., T.J., D.R.D., T.S., J.A.S., and C.B.; validation: R.S. and W.S.K.; formal analysis: R.S. and W.S.K.; investigation: R.S. and W.S.K.; resources: C.L.Y., R.S., W.S.K., and T.J.; writing—original draft preparation: C.L.Y., P.C.T., C.L., M.C., T.J., R.S., W.H.S., A.G., and T.S.; writing—review and editing: all authors; visualization: W.S.K., T.J., C.L.Y., C.L., and P.C.T.; supervision: C.L.Y. and C.L; project administration: C.L.Y.; funding acquisition: C.L.Y., C.L., P.C.T., and M.C. All authors have read and agreed to the published version of the manuscript.

**Funding:** This research received no external funding. Internal NASA funding was awarded through competed proposals to Internal Research and Development (IRAD) and LaRC Center Transformation opportunities. Additional support was provided through the LaRC Science Directorate.

**Acknowledgments:** The authors would like to thank the following individuals from NASA LaRC for their support of this project: Laura Rogers for helping to secure transformation funding; Timothy Marvel for creating conceptual artwork; David Beals for mentorship and meaningful conversations; Rebecca Bales for proposal development support; Susan Johnston for help with early project management; Sandra Chellis, Susan Thomas, Lisa Yoakum, Sherry Monk, and Josephine Sawyer for contracts and procurement support; Amber Richards for team meeting logistics; Drew Hope for providing the opportunity to kick-off the design session in 2016; Chris Edwards for making available the Maker's Space facility to host the team meeting in 2019; Norman Loeb, Allen Larar, David Beals, and Emma Brand for reviewing a draft of this document.

**Conflicts of Interest:** The authors declare no conflict of interest. The funders had no role in the design of the study; in the collection, analyses, or interpretation of data; in the writing of the manuscript; or in the decision to publish the results.

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
