# Peer review of "Trutinor: A Conceptual Study for a Next-Generation Earth Radiant Energy Instrument"

_remotesensing, doi:10.3390/rs12203281_

Round 1
Reviewer 1 Report
Minor revision
The paper is clear and language is fine.
Following changes (minor revision) are requested:
- Figure 3A please insert in figure caption or in the text what are d, Dd, and
- The description of the instrument concept could be improved by adding a more extensive description of figure 5. For example, by adding in figure and description more details on FPA.
- Remove reference "[25]" from conclusions.
Author Response
The paper is clear and language is fine.
We appreciate your review of our manuscript.
Following changes (minor revision) are requested:
- Figure 3A please insert in figure caption or in the text what are d, Dd, and
The figure caption has been edited to include the definitions of beta and delta.
- The description of the instrument concept could be improved by adding a more extensive description of figure 5. For example, by adding in figure and description more details on FPA.
Text has been improved (lines 314 - 321). The FPA selected for concept prototyping is identified in the Trutinor Breadboard section.
- Remove reference "[25]" from conclusions.
We have removed this reference in the conclusions.
Reviewer 2 Report
Review of "Trutinor: A Conceptual Study for a Next-Generation Earth Radiant Energy Instrument" by Young et al., 2020.
The manuscript is a conceptual study for a possible instrument development to take over the current NASA CERES instruments to continuously measure the Earth Radiation Budget without temporal gaps.
The instrument is well described, and the whole manuscript makes sense for a conceptual study.
Nevertheless, some minor changes are required before publication. Some parts suffer from hasty writing and should be extended, i.e. lines 103-106 (pag. 3) describe CERES measurements. Many details are left to the reader. A more detailed description would for sure improve clarity.
I would also suggest to introduce somewhere in the manuscript a table summarizing the principal differences (also economical) between CERESE instruments and Trutinor.
Some figures look stretched with fonts difficult to read.
Author Response
Review of "Trutinor: A Conceptual Study for a Next-Generation Earth Radiant Energy Instrument" by Young et al., 2020.
The manuscript is a conceptual study for a possible instrument development to take over the current NASA CERES instruments to continuously measure the Earth Radiation Budget without temporal gaps.
The instrument is well described, and the whole manuscript makes sense for a conceptual study.
Nevertheless, some minor changes are required before publication. Some parts suffer from hasty writing and should be extended, i.e. lines 103-106 (pag. 3) describe CERES measurements. Many details are left to the reader. A more detailed description would for sure improve clarity.
Many thanks for your review of our manuscript.
The CERES instrument approach and measurements are described in more detail in Section 3 “State of ERB technology”. We added a reference to this Section on lines 104-105.
I would also suggest to introduce somewhere in the manuscript a table summarizing the principal differences (also economical) between CERES instruments and Trutinor.
We agree, having such a table is helpful when comparing instruments with mature “flight” designs. In our case, the TRL (technology readiness level) of Trutinor is too low to reliably estimate key parameters such as size, mass, power, and resulting costs. In our opinion, Trutinor’s measurement parameters are sufficiently provided in the text.
Some figures look stretched with fonts difficult to read.
Thank you for pointing this out. At the final proofing of the manuscript, we will make sure all text in the Figures is readable.
Reviewer 3 Report
A nice study on Earth Radiant Energy ........
Author Response
A nice study on Earth Radiant Energy ........
Thank you very much for reviewing our manuscript!
Reviewer 4 Report
Title: “Trutinor: A Conceptual Study for a Next-Generation Earth Radiant Energy Instrument”
Young et al.
This manuscript describes the conceptual design of a follow-on instrument to CERES, which is required to reduce and eliminate the potential for a gap in the global ERB record. The Trutinor instrument is compact and relatively inexpensive, will be high in accuracy, and will have a smaller footprint than CERES, allowing for better science and ease of comparison with other ERB sensors.
The manuscript is extremely well written and organized. The authors have clearly taken their time with writing such a comprehensive overview of this new sensor. For this, I believe the manuscript is ready for publication, however, I have just a few questions about Trutinor that I would like to have answered prior to publication. The authors should consider the below questions (they do not necessarily need to be answered in the text, but this reviewer would appreciate a response):
- How many Trutinor sensors do you expect to be needed to cover the anticipated 50-year temporal coverage?
- Do you have a sense for that the requirements for a “collocated” imager? You mention a 3-min lag, but are there any other requirements (e.g., specific channels)? Will Trutinor still be scientifically relevant if the specific imager it is collocated with fails? Will you be seeking multiple platform opportunities? For example, if the opportunity presents itself to where you can fly on a major satellite platform, will you take it? Or are you more focused on smaller platforms?
- With regards to SW calibration, what if the lunar model is not completed prior to launch? You note that these models are not yet completed. What challenges do you anticipate (or what backup plans do you have) if these calibration models are not successfully built by launch time? Are there options for post calibration?
Line 108: consider including other studies, such as Stanfield et al. 2014; Dolinar et al. 2015; Wild et al. 2015
Author Response
This manuscript describes the conceptual design of a follow-on instrument to CERES, which is required to reduce and eliminate the potential for a gap in the global ERB record. The Trutinor instrument is compact and relatively inexpensive, will be high in accuracy, and will have a smaller footprint than CERES, allowing for better science and ease of comparison with other ERB sensors.
The manuscript is extremely well written and organized. The authors have clearly taken their time with writing such a comprehensive overview of this new sensor. For this, I believe the manuscript is ready for publication, however, I have just a few questions about Trutinor that I would like to have answered prior to publication. The authors should consider the below questions (they do not necessarily need to be answered in the text, but this reviewer would appreciate a response):
We thank the reviewer for their comments, and we have responded below to their questions. Since our responses have the nature of a general discussion, we did not include them in the revised manuscript.
- How many Trutinor sensors do you expect to be needed to cover the anticipated 50-year temporal coverage?
To our knowledge, the leading nano- and micro-satellite vendors design systems with a 5-year lifetime in a nominal 500 km sun synchronous orbit. As of today, there are no data points for such long performance, but we expect that in the near future this would be realistic. With this assumption, we would need 10 sensors to cover a 50-year time period. However, this would depend on the future ERB architecture. For example, if we wanted to have one sensor in a morning orbit and one in an afternoon orbit, we would need 20 sensors to cover a 50-year time span.
- Do you have a sense for that the requirements for a “collocated” imager? You mention a 3-min lag, but are there any other requirements (e.g., specific channels)? Will Trutinor still be scientifically relevant if the specific imager it is collocated with fails? Will you be seeking multiple platform opportunities? For example, if the opportunity presents itself to where you can fly on a major satellite platform, will you take it? Or are you more focused on smaller platforms?
These are all very good questions. (1) In our opinion, the requirements for an ERB imager (spectral bands, spatial resolution) can be relaxed if we compare to those of general imagers like MODIS and VIIRS. This is because the imager information is used primarily for ADM scene identification. For example, not all imager bands are used in the current CERES algorithms, and a spatial resolution of 1 km would be sufficient. (2) If the near-coincident imager fails, Trutinor’s fluxes would have larger instantaneous uncertainty, but still good average fluxes (same as CERES). The CERES ERB project has AI-based algorithms for the situation when imager data is not available (Lukashin and Loeb, 2004). (3) The conceptual idea for Trutinor is to design an instrument for micro-satellite platforms that is compact, with low mass and power, and uses microsat ability to point for calibration and ADM data collection (no heavy, power consuming gimbals). Accommodation on large platforms would require a gimbal, therefore we are focused on smaller platforms.
- With regards to SW calibration, what if the lunar model is not completed prior to launch? You note that these models are not yet completed. What challenges do you anticipate (or what backup plans do you have) if these calibration models are not successfully built by launch time? Are there options for post calibration?
This is correct, the existing ROLO (lunar) calibration model is not accurate enough for ERB, and it is not extended to the SW broadband yet. However, the Trutinor-like instrument deployment is probably 7-10 years away. Meanwhile, there are a few current projects that address the improvement needed in lunar calibration: CLARREO Pathfinder (launch in 2023), ARCSTONE (ESTO IIP), Ground- and Air-LUSI (ground and airborne lunar measurements by NIST and UMBC). We hope that in 7-10 years we will have an accurate lunar calibration model extended to the SW broadband. Also, if this model is available later, it would still be applicable (lunar calibration is literally timeless). The backup plan could be using the Sun as a calibration source, however signal reduction needs to be implemented by using diffusers and/or apertures.
Line 108: consider including other studies, such as Stanfield et al. 2014; Dolinar et al. 2015; Wild et al. 2015.
Thank you for these suggestions. We have added these additional references to the text.